# Design and Testing of a Non-Contact MEMS Voltage Sensor Based on Single-Crystal Silicon Piezoresistive Effect

**DOI:** 10.3390/mi13040619

**Published:** 2022-04-15

**Authors:** Jiachen Li, Jun Liu, Chunrong Peng, Xiangming Liu, Zhengwei Wu, Fengjie Zheng

**Affiliations:** 1State Key Laboratory of Transducer Technology, Aerospace Information Research Institute, Chinese Academy of Sciences, Beijing 100190, China; lijiachen19@mails.ucas.ac.cn (J.L.); sushi_1220@163.com (J.L.); liuxiangming18@mails.ucas.ac.cn (X.L.); zwwu@mail.ie.ac.cn (Z.W.); fjiezheng@163.com (F.Z.); 2University of Chinese Academy of Sciences, Beijing 100049, China

**Keywords:** non-contact voltage measurement, piezoresistive microstructure, low power, MEMS

## Abstract

The paper presents a novel non-contact microelectromechanical systems (MEMS) voltage sensor based on the piezoresistive effect of single-crystal silicon. The novelty of the proposed sensor design lies in the implementation of unique single-crystal silicon piezoresistive beams for voltage measurement. The sensitive structure of the sensor produces electrostatic force deformation due to the measured voltage, resulting in the resistance change of single-crystal silicon piezoresistive beams which support a vibrating diaphragm. The voltage can be measured by sensing the resistance change. Moreover, the sensor does not need an additional driving signal and has lower power consumption. The prototype of the sensor was fabricated using an SOI micromachining process. The piezoresistive characteristics of the sensor and the corresponding output response relationship were analyzed through theoretical analysis and finite element simulation. The voltage response characteristics of the sensor were achieved at power frequencies from 50 Hz to 1000 Hz in the paper. The experimental results showed that they were in good agreement with simulations results with the theoretical model and obtained good response characteristics. The sensor has demonstrated that the minimum detectable voltages were 1 V for AC voltages at frequencies from 50 Hz to 300 Hz and 0.5 V for AC voltages at frequencies from 400 Hz to 1000 Hz, respectively. Moreover, the linearities of the sensor were 3.4% and 0.93% in the voltage measurement range of 900–1200 V at the power frequency of 50 Hz and in the voltage measurement range of 400–1200 V at the frequency of 200 Hz, respectively.

## 1. Introduction

Voltage sensors are widely used in various aspects of power systems, such as transmission, substation, and distribution, to provide accurate and reliable signals for metering, measurement, and control and relay protection devices. Presently, commonly used voltage sensors mainly include contact and non-contact of which contact voltage sensors need to be electrically connected to the power system, which lacks electrical isolation and has security risks, while non-contact voltage sensors cater to the development trend of modern power systems for voltage measurement, with the advantages of more secure and reliable, simple equipment and easy testing [1,2].

In recent years, there have been many reports on non-contact voltage measurement. For optical non-contact voltage sensors, which are mostly based on the electro-optical effect and optical fiber [3,4,5,6], although the size of the optical sensor is small and the insulation is good, it is difficult to process optical components, which are high in cost and low in long-term stability and reliability. For non-contact voltage sensors based on the coupling capacitor [7], which realize the non-contact measurement of voltage by connecting the coupling capacitor to the impedance circuit. These sensors are easy to test, safe and reliable but have poor interference immunity and are susceptible to circuitry and parasitic capacitance. For non-contact voltage sensors based on the measuring principle of the D-dot probe [8], the insulation of the sensor is achieved, and the size of the sensor is smaller; however, the measurement of such sensors is susceptible to environmental influences, and the anti-interference ability and stability need to be improved. For non-contact voltage sensors based on the voltage divider principle, which use the resistive voltage divider, capacitive voltage divider, or resistive-capacitive voltage divider principle to achieve non-contact measurement of voltage, the most significant problem they have is their vulnerability to external ambient temperature and external stray capacitance.

With the development of the microelectromechanical system (MEMS), some non-contact MEMS voltage sensors are reported due to their small size, low power consumption, easy mass production, and integration [9,10,11]. It is very suitable for large-scale applications such as smart grids and the Internet of things in power systems. These types of sensors have a wide range of structures, but all of them mainly contain electrode-sensitive structures and drive structures. Presently, these sensors are mainly driven by electrostatic drive [12,13], thermal drive [14], and piezoelectric drive [15,16], which leads to high power consumption and complex structure. In summary, there are many shortcomings in the research of non-contact voltage sensors at present, so there is an urgent need to research new miniaturized, low-power, high-performance non-contact voltage sensors.

In this paper, a novel non-contact MEMS voltage sensor based on the piezoresistive effect of a single-crystal silicon structure was proposed. Compared with other voltage sensors, the proposed sensor adopted a unique piezoresistive single-crystal silicon microstructure design, which did not need an additional driving signal and had lower power consumption.

## 2. Principle and Design

The structure of the MEMS voltage sensor is shown in Figure 1. The sensor consisted of a vibrating diaphragm, supporting beams, and piezoresistive beams. Piezoresistive beams had a straight beam microstructure, and they were connected to a vibrating diaphragm through supporting beams, which was a folded beam structure to reduce the system stiffness. The holes in the membrane of the MEMS device were designed to reduce air damping and facilitate the release of the sensor structure.

The operating principle of the MEMS voltage sensor is shown in Figure 2. When the sensor is placed at a distance, d, from the voltage source, the sensor converts the voltage between the voltage source and the diaphragm of the sensor into an electrostatic force. The vibrating diaphragm is displaced vertically because of the electrostatic force, which causes the deflection of the piezoresistive beams. Therefore, the resistance value of the piezoresistive beam changes due to the piezoresistive effect [17]. In this paper, the measured voltage on the voltage source is obtained by measuring the change in resistance value.

## 3. Theory and Simulation

The electrostatic force pulling on the diaphragm due to the voltage between the voltage source and the diaphragm of the sensor is given by [18]:(1)$$F_{e}=\frac{\varepsilon_{0}\cdot \varepsilon_{r}\cdot A\cdot V_{s}{}^{2}}{2\cdot d^{2}},$$
where d is the distance separating the diaphragm and the voltage source; *A* is the surface area of the diaphragm; $V_{s}$ is the voltage on the nearby voltage source; $\varepsilon_{0}$ and $\varepsilon_{r}$ are the vacuum permittivity and the relative permittivity, respectively. When the electrostatic force is loaded at the diaphragm, the concentrated force on the piezoresistive beam is one-fourth of the electrostatic force, which can be analyzed from Figure 1. Therefore, the loading force can be described as:(2)$$F=\frac{F_{e}}{4}=\frac{\varepsilon_{0}\cdot \varepsilon_{r}\cdot A\cdot V_{s}{}^{2}}{8\cdot d^{2}}.$$

The piezoresistive beam can be regarded as a doubly clamped beam. A central concentration force is loaded at the piezoresistive beam due to the electrostatic force, which leads to the deflection of the piezoresistive beams. The origin of the piezoresistive conversion can be illustrated by considering the vibration mode shape of the beam, as shown in Figure 3.

Based on the deflection profile of the doubly clamped beam vibrating in its fundamental mode as described by the Euler beam equation [19], looking at the position x as shown in Figure 4, the displacement function of the beam is written as:(3)$$w\left(x\right)=\frac{\frac{FL}{16}x^{2}-\frac{F}{12}x^{3}}{EI},\left(0\leq x\leq \frac{L}{2}\right),$$
where L is the length of the beam at rest, E is Young’s modulus of the beam, I is the moment of inertia of the beam, and F is the loading force at the middle of the beam as shown in Equation (2). The aggregate axial strain induced by the deflection of the beam is given by [17]:(4)$$\varepsilon_{x}=\frac{1}{2L}\int_{0}^{L}{\left(\frac{\partial w}{\partial x}\right)}^{2}dx,$$
where L is the length of the beam at rest, and w is the deflection function of the beam, as shown in Equation (3). Therefore, according to Equations (2) and (3), the aggregate axial strain induced by the deflection of the beam, Equation (4) can be written as:(5)$$\varepsilon_{x}=a_{0}V_{s}{}^{4},$$
where $a_{0}$ is a numerically determined constant. For the piezoresistive beam, its resistance varies, when there is an axial strain. The variation can be written as [20]:(6)$$\frac{\Delta R}{R}=\pi_{l}\sigma_{l}=\pi_{l}E\varepsilon_{x},$$
where ΔR is the resistance variation; R is the original resistance value; $\pi_{l}$ is the longitudinal piezoresistive coefficient, which is constant for a fabricated piezoresistive beam; $\sigma_{l}$ and $\varepsilon_{x}$ are the axial stress and the axial strain, respectively; E is Young’s modulus of the beam. According to Equations (5) and (6), the resistance variation of the piezoresistive beam can be written as:(7)$$\frac{\Delta R}{R}=\pi_{l}Ea_{0}V_{s}{}^{4}=c_{0}V_{s}{}^{4},$$
where $c_{0}$ is also a constant. It can be seen from Equation (7) that the change in the resistance value of the piezoresistive beam is proportional to the fourth power of the measured voltage. In addition, when the measured voltage is an AC voltage $V_{s}=V_{m}\;\cos\left(wt\right)$, Equation (7) can be written as:(8)$$\frac{\Delta R}{R}=C_{0}V_{m}{}^{4}\left(\frac{3}{8}+\cos\left(2wt\right)+\frac{\cos\left(4wt\right)}{8}\right).$$

According to Equation (8), the change in the resistance value of the piezoresistive beam contains a DC component, the second harmonic component, and the fourth harmonic component. In this paper, in order to obtain higher sensitivity and reduce noise, we extracted the signal of the second-harmonic component by filtering to realize the piezoresistive detection, which is proportional to the fourth power of the measured voltage.

To better illustrate the relationship between the resistance change of the piezoresistive beam and the measured voltage, according to Equation (6), the relationship between the axial strain of the piezoresistive beam and the amplitude of the measured voltage was analyzed by using COMSOL finite element simulation software. The simulation model was established as shown in Figure 1, applying the measured voltage above the grounding diaphragm. The boundary conditions were applied to the model, and the simulation results were obtained after meshing and calculation, as shown in Figure 5.

It can be seen from Figure 5 that the axial strain of the piezoresistive beam was proportional to the fourth power of the measured voltage. In addition, according to Equation (6), the change in the resistance value of the piezoresistive beam is linear with the axial strain of the piezoresistive beam; therefore, the change in the resistance value of the piezoresistive beam was proportional to the fourth power of the measured voltage, which was in good agreement with the theoretical model.

To the design of the parameters of the sensor structure, according to Equations (2)–(4) and (7), the vibrating diaphragm area A (the side length of the vibrating diaphragm area $L_{d}$) and the length of the piezoresistive beam L have the greatest effects on the resistance variation of the piezoresistive beam. Therefore, Figure 6 shows the finite element simulation results, which is the relationship between the axial strain of the piezoresistive beam and the parameters of the sensor structure.

As can be seen from Figure 6, under the same conditions, the larger the length of the piezoresistive beam L and the side length of the vibrating diaphragm $L_{d}$, the greater the axial strain of the piezoresistive beam and the greater the resistance variation of the piezoresistive beam. Therefore, based on the comprehensive consideration of the sensor structure characteristics analysis, the MEMS manufacturing process rules, and its key dimensional processing constraints, the key parameters of the sensor structure design are shown in Table 1.

## 4. Sensor Fabrication

The device was fabricated in a commercial SOIMUMPS process. The SOIMUMPs process is a simple 4-mask-level SOI patterning and etching process. The piezoresistive device was developed as shown in Figure 7.

The sensor chip was fabricated as below:(a)Silicon doping: The top surface of the silicon layer was doped by depositing a phosphor silicate glass (PSG) layer and annealing at 1050 °C for 1 h in argon. The PSG layer was then removed via wet chemical etching;(b)Pad metal liftoff: The padded metal was deposited over the device layer by e-beam evaporation;(c)Silicon patterning: Silicon was lithographically patterned with a mask and etched using deep reactive ion etching (DRIE);(d)Substrate patterning: A frontside protection material was applied to the top surface of the patterned silicon layer. The wafers were then reversed, and the substrate layer was lithographically patterned from the bottom side using a mask. This pattern was then etched into the bottom side oxide layer using reactive ion etching (RIE);(e)Protection layer removal: The frontside protection material was then stripped using a dry etching process.

The image of the sensor is shown in Figure 8.

## 5. Fabrication Experiment Setup

To realize the performance test of the sensor, the test system platform of the sensor was designed. The measurement principle of the sensor is shown in Figure 9. The voltage $V_{d}$ and $-V_{d}$ were loaded at both ends of the piezoresistive beam, which caused the vibrating diaphragm grounding. When the sensor was placed at a distance from the measured voltage source, the vibrating diaphragm was displaced vertically due to the electrostatic force, which led to the deflection of the piezoresistive beam. Therefore, the dynamic current in the piezoresistive beam was changed, and the two piezoresistive signals through the isolation capacitor were differentially amplified by AD620. When the resistance value of the piezoresistive beam changes, the dynamic current in the piezoresistive beam can be described as:(9)$$i\approx \frac{V_{d}}{R}\left(\frac{\Delta R_{r}}{R_{r}}\right)\;,$$
where $R=R_{r}+2R_{0}$; $V_{d}$ is the supply voltage of the circuit; $\Delta R_{r}$ is the change in the resistance value of the piezoresistive beam; $R_{r}$ is the original resistance value of the piezoresistive beam; $R_{0}$ is the series resistor. Therefore, according to Equation (9), the output voltage at both ends of the piezoresistive beam can be written as:(10)$$V_{i+}\approx i\cdot \frac{R_{r}}{2}=\frac{V_{d}}{R}\left(\frac{\Delta R_{r}}{R_{r}}\right)\cdot \frac{R_{r}}{2},$$
(11)$$V_{i-}\approx -i\cdot \frac{R_{r}}{2}=-\frac{V_{d}}{R}\left(\frac{\Delta R_{r}}{R_{r}}\right)\cdot \frac{R_{r}}{2}.$$

From the above equations, the output voltage is proportional to the change in resistance of the piezoresistive. According to Equation (8), the output voltage contains a DC component, the second-harmonic component, and the fourth harmonic component. Therefore, a lock-in amplifier was used to measure the second-harmonic component to obtain higher sensitivity and reduce noise. The amplitude of output voltage is proportional to the fourth power of the measured voltage.

The test system was composed of a power section, the measured voltage power, a lock-in amplifier, a PC, and the measurement circuit, as shown in Figure 10. The power section was used to provide a DC voltage to the circuit, and the voltage $V_{d}$ was 5 V. The positive terminal of the measured voltage power was connected to a metal pole plate, and the negative terminal was common ground with the circuit. In order to verify the performance of the sensor, the sensor was encapsulated in ceramic and inverted on the metal plate, to which the measured high voltage source was applied. A high-voltage source was a high-precision-power YOKOGAWA 2558A instrument. The high-precision-power YOKOGAWA 2558A was used as the measured voltage power to provide AC voltages in the voltage range of 0–1200 V at the frequency range of 40–1000 Hz. Therefore, the measured voltage caused the deflection of the piezoresistive beam because of the electrostatic force between the vibrating diaphragm and the measured voltage. An HF2LI lock-in amplifier was used to measure the second-harmonic component of the output signals.

## 6. Results

### 6.1. Voltage and Frequency Responses

The voltage response curves of the sensor at different frequencies and the frequency response curves of the sensor at different voltages can be obtained by changing the frequency and amplitude of the measured voltage power, as shown in Figure 11.

It can be seen from Figure 11a that there was a non-linear relationship between the sensor output voltage and the measured voltage, and the sensor output voltage increased with the measured voltage. To explain the relationship between the output voltage and the measured voltage more clearly, the voltage response curves of the sensor at the frequencies of 50 Hz, 200 Hz, 500 Hz, and 1000 Hz are curve-fitted, as shown in Figure 12.

It can be concluded from Figure 12 that the output voltage was proportional to the fourth power of the measured voltage, which is in good agreement with the theoretical model. In addition, there was also a non-linear relationship between the sensor output voltage and the frequency of the measured voltage, as shown in Figure 11b, and the sensor output voltage increased with the frequency of the measured voltage in the measurement frequency range.

### 6.2. Linearity and Sensitivity

According to Figure 11, the output voltage was linearly proportional to the voltage measurement range of 900–1200 V at the frequency of 50 Hz, the voltage measurement range of 400–1200 V at the frequency of 200 Hz, the voltage measurement range of 300–1200 V at the frequency of 500 Hz, and the voltage measurement range of 200–1200 V at the frequency of 1000 Hz separately, as shown in Figure 13.

According to Figure 13, the data showed a good linear relationship between the sensor output voltage and the measured voltage. The linearity and sensitivity values at different frequencies can be calculated as shown in Table 2. Moreover, the lowest voltage amplitudes that could be accurately measured were 1 V from 50 Hz to 300 Hz and 0.5 V from 400 Hz to 1000 Hz, as shown in Table 2.

Figure 13 and Table 2 show that the sensor had a good linear response to the voltage amplitude and can be used to measure AC voltages at power frequencies from 50 Hz to 1000 Hz in this paper.

Compared with other MEMS voltage sensors [21,22,23] as shown in Table 3, the voltage sensor proposed in this paper did not need any driving structures and any additional driving signals, which realized the low-power measurement of the sensor. Moreover, the test circuit of the sensor was simple, and the static power consumption of the circuit depended on the series resistor $R_{0}$. In addition, when the series resistor $R_{0}$ was 100 kΩ, the sensor consumption including the test circuit was on the order of milliwatts or lower.

## 7. Conclusions

In this paper, a novel MEMS non-contact voltage sensor based on the piezoresistive effect of a single-crystal silicon structure was demonstrated. The sensor achieved low power consumption measurement without any driving structure and any additional drive signals. In addition, the power dissipation of the test circuit was on the order of milliwatts or lower. The entire single-crystal silicon beam was used as a piezoresistive element to improve the sensitivity of the sensor and simplify the process. Based on the SOI MEMS process, the device prototype was fabricated. Experimental tests showed that the sensor output response characteristics were consistent with theoretical analysis and the sensor has obtained fine linearity and sensitivity. The lowest voltage amplitudes experimentally measured were 1 V at frequencies from 50 Hz to 300 Hz and 0.5 V at frequencies from 400 Hz to 1000 Hz. The linearities of the sensor were 3.4% and 0.93% in the voltage measurement range of 900–1200 V at the power frequency of 50 Hz and the voltage measurement range of 400–1200 V at the power frequency of 200 Hz, respectively. In the future, we will further carry out application verification for practical application scenarios, such as power transmission, transformation, and distribution, and study the anti-interference characteristics of the sensor. In addition, for low-voltage measurements, such as below 220 V, the sensor will be further optimized to improve the measurement sensitivity.

## Figures and Tables

**Figure 1 micromachines-13-00619-f001:**
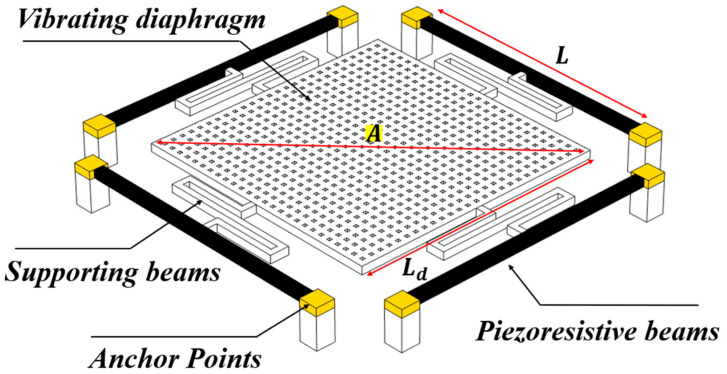
The structure of the sensor.

**Figure 2 micromachines-13-00619-f002:**
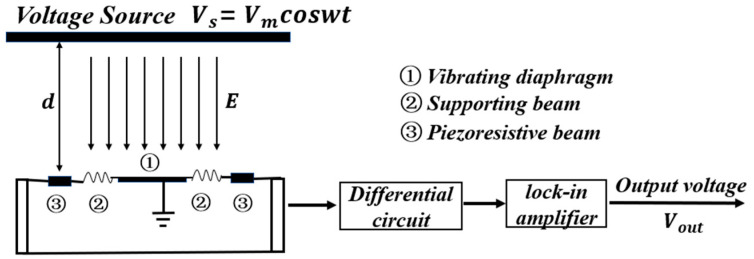
The operating principle of the sensor.

**Figure 3 micromachines-13-00619-f003:**
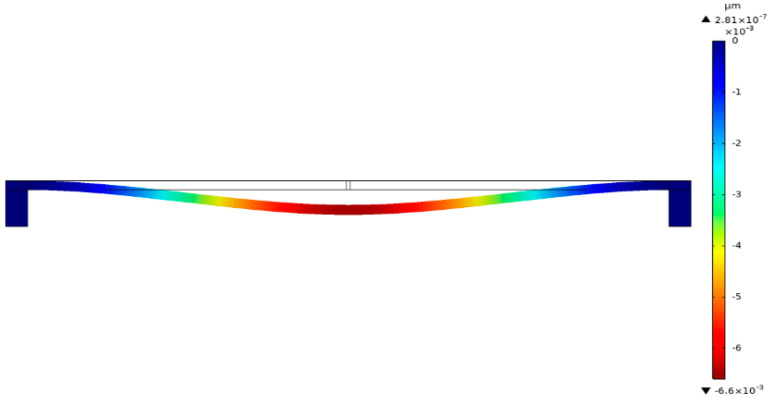
The deflection of a center−loaded doubly clamped beam.

**Figure 4 micromachines-13-00619-f004:**
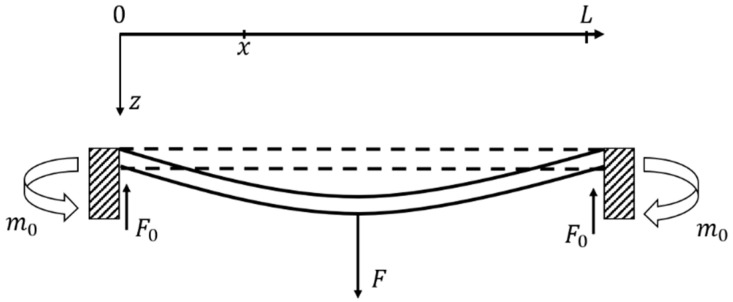
Doubly clamped beam (bridge) and its coordinates.

**Figure 5 micromachines-13-00619-f005:**
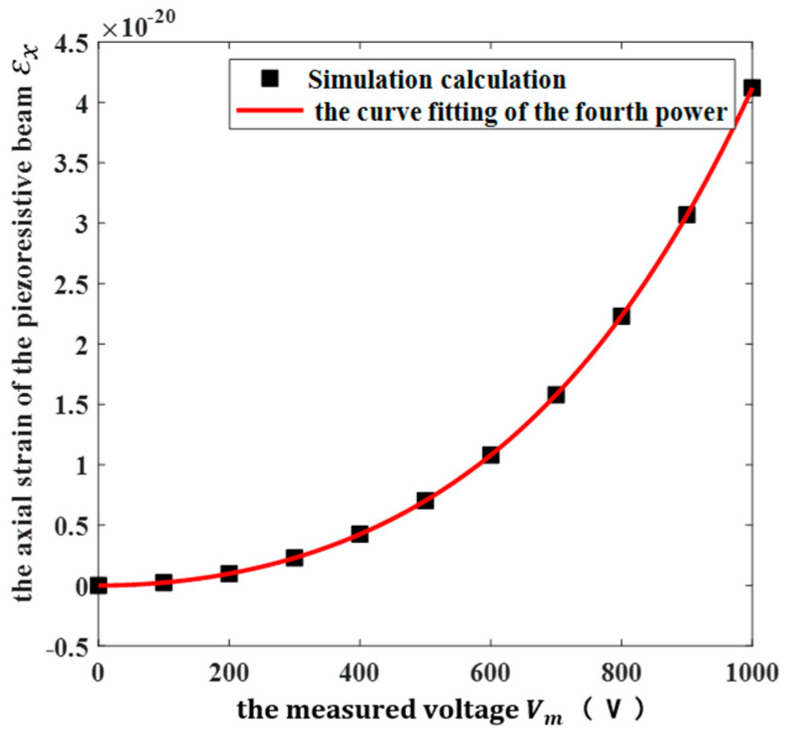
The simulation results between the axial strain of the piezoresistive beam $\varepsilon_{x}$ and the amplitude of the measured voltage $V_{m}$.

**Figure 6 micromachines-13-00619-f006:**
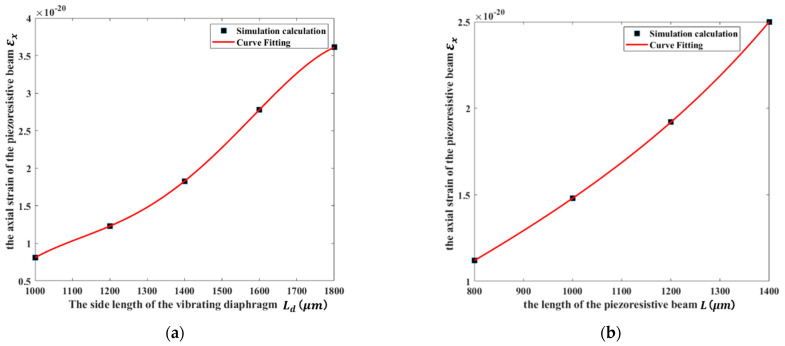
The simulation results between the axial strain of the piezoresistive beam $\varepsilon_{x}$ and the parameters of the sensor structure: (**a**) the side length $L_{d}$ of the vibrating diaphragm; (**b**) the length L of the piezoresistive beam.

**Figure 7 micromachines-13-00619-f007:**
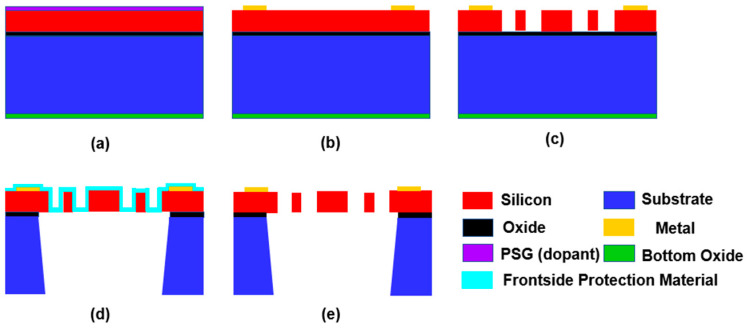
The chip fabrication flow.

**Figure 8 micromachines-13-00619-f008:**
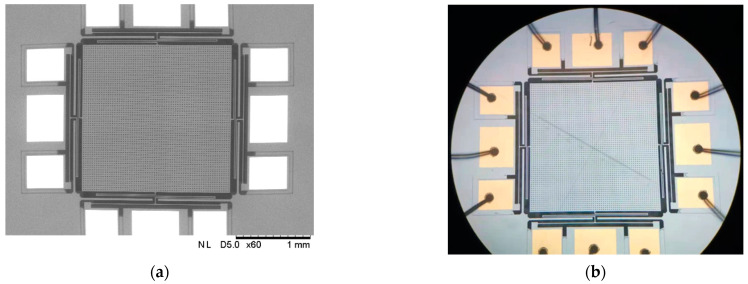
(**a**) The SEM image of the sensor; (**b**) the picture of sensor chip under a microscope.

**Figure 9 micromachines-13-00619-f009:**
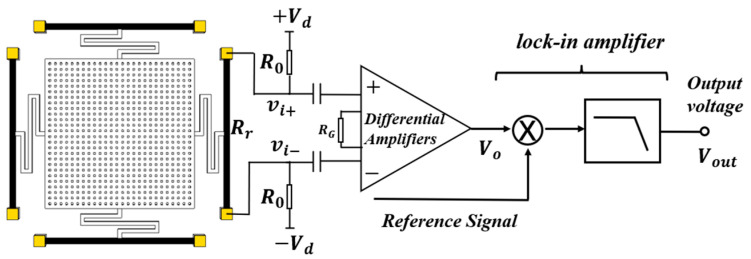
The measurement principle of the sensor.

**Figure 10 micromachines-13-00619-f010:**
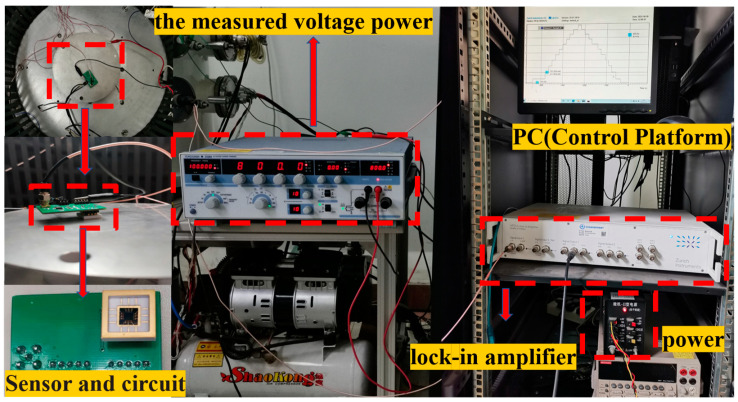
Photo of the experimental setup.

**Figure 11 micromachines-13-00619-f011:**
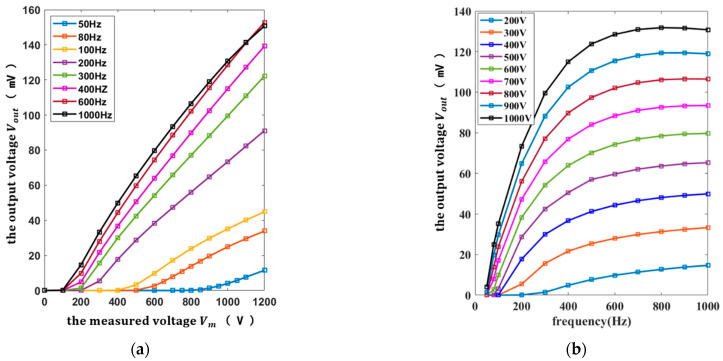
The voltage response curves of the sensor at different frequencies and the frequency response curves of the sensor at different voltages: (**a**) the voltage response curve of the sensor; (**b**) the frequency response curve of the sensor.

**Figure 12 micromachines-13-00619-f012:**
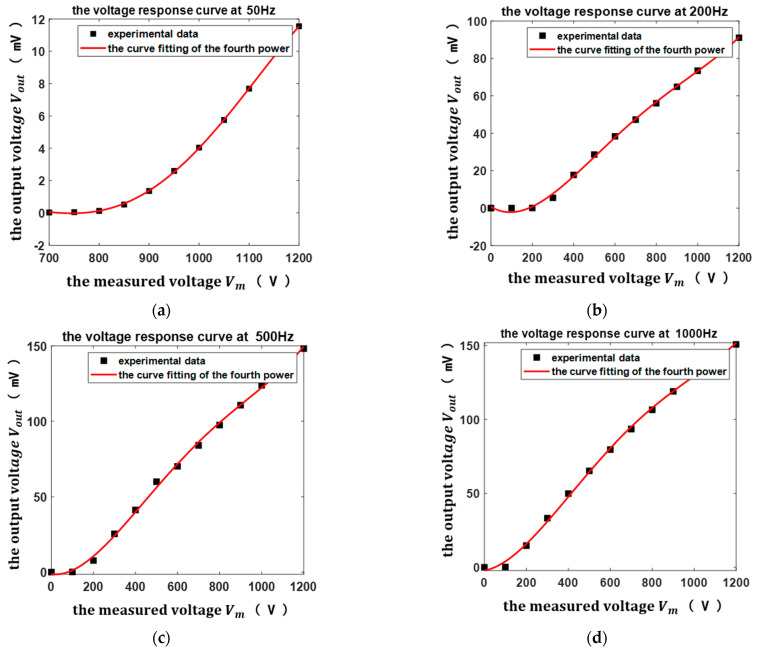
The voltage response curves of the sensor curve-fitted at the frequencies: (**a**) 50 Hz; (**b**) 200 Hz; (**c**) 500 Hz; (**d**) 1000 Hz.

**Figure 13 micromachines-13-00619-f013:**
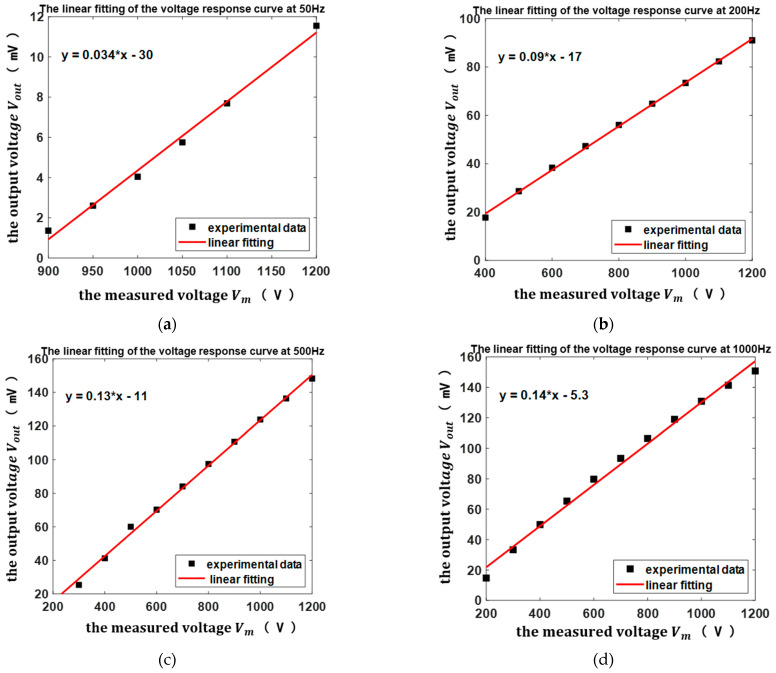
The linear fittings of the sensor voltage response curves at frequencies: (**a**) 50 Hz; (**b**) 200 Hz; (**c**) 500 Hz; (**d**) 1000 Hz.

**Table 1 micromachines-13-00619-t001:** The key parameters of the sensor.

Key Parameters	Value
the side length of the vibrating diaphragm $L_{d}$	2000 μm
the length of the piezoresistive beam L	1750 μm
the width of the piezoresistive beam $W_{p}$	10 μm
the thickness of the piezoresistive beam t	25 μm
the width of the supporting beam $W_{s}$	10 μm
the single section length of the supporting beam $L_{s}$	800 μm

**Table 2 micromachines-13-00619-t002:** The linearity, sensitivity, and resolution of the sensor.

Frequency (Hz)	Linearity	Sensitivity (mV/V)	Resolution (V)
50	3.4%	0.034	1
200	0.93%	0.09	1
500	1.62%	0.13	0.5
1000	3.87%	0.14	0.5

**Table 3 micromachines-13-00619-t003:** The comparison with other microelectromechanic systems (MEMS) voltage sensors.

Source	Driving Structure	Driving Signals
[21]	thermal drive	a driving voltage of 75 mV (vacuum environment)
[22]	electrostatic drive	a DC bias voltage of 20 V and an AC voltage amplitude of 1 V
[23]	electrostatic drive	a DC bias voltage of 8 V and an AC voltage of 10 V (peak-to-peak)
this work	no driving structure	no driving voltage

## Data Availability

Not applicable.

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
