# Peer review of "Design and Testing of a Non-Contact MEMS Voltage Sensor Based on Single-Crystal Silicon Piezoresistive Effect"

_micromachines, 2022, doi:10.3390/mi13040619_

Round 1
Reviewer 1 Report
In this paper, the authors presented a voltage sensor based on single crystal silicon piezoresistive effect. And they have systematic investigation of the sensor from theoretical simulating to device performance. However, there still are some comments for the authors before this paper be published.
- It is suggested to compared the sensitivity of the sensor presented in this paper with the other reported non-contact voltage sensors.
- It is better to give more details about the sensor performance under the voltage below 200 V. Furthermore, can the sensitivity of the sensors be improved with detecting the voltage below 200 V? Since the voltage around 200 V are commonly used.
- It is better to add a scale bar in Figure 8.
- The authors emphasized that this sensor has low power consumption, it is suggested to compare the power consumption of this sensor with others to highlight its merit of low power consumption.
Author Response
Dear Reviewer,
Thanks for your valuable comments for the manuscript entitled “Design and Testing of a Novel Non-contact MEMS Voltage Sensor Based on Single-Crystal Silicon Piezoresistive Effect”. We have tried our best to revise the manuscript. Revised portions are marked using the “Track Changes” function in the revised manuscript. And the point to point responds to your comments are listed as following:
Point 1: It is suggested to compared the sensitivity of the sensor presented in this paper with the other reported non-contact voltage sensors.
Response 1: Thank you very much for your suggestion. The sensitivity of the sensor depends on the design of the sensor and the gain of the amplification circuit. We also try to compare the sensitivity with other reported non-contact voltage sensors. However, other reported non-contact voltage sensors have different amplification circuits and magnification, therefore their comparison is not consistent and their sensitivities cannot be compared together. Moreover, the focus of this paper is to propose a novel non-contact MEMS voltage sensor based on single-crystal silicon piezoresistive effect rather than a high-sensitivity sensor.
Point 2: It is better to give more details about the sensor performance under the voltage below 200 V. Furthermore, can the sensitivity of the sensors be improved with detecting the voltage below 200 V? Since the voltage around 200 V are commonly used.
Response 2: Thank you very much for your suggestion. In household electricity and other low-voltage application scenarios, about 200V is indeed commonly used. However, the main purpose of this paper is to verify the sensitivity mechanism of the new non-contact voltage sensor with single-crystal silicon piezoresistive effect and its response characteristics in a larger voltage range including 200V. Because in scenarios such as power transmission, transformation, and distribution, the voltage is usually greater than 200V, but at present, our test is limited to standard voltage sources. In the future, we will further optimize to improve the sensitivity of the sensor to satisfy detecting the voltage below 200 V.
Point 3: It is better to add a scale bar in Figure 8.
Response 3: Thank you very much for your suggestion. In the revised manuscript, we add a SEM image of the sensor.
Point 4: The authors emphasized that this sensor has low power consumption, it is suggested to compare the power consumption of this sensor with others to highlight its merit of low power consumption.
Response 4: Thank you very much for your suggestion. The voltage sensor proposed in this paper is designed with a MEMS structure. And this sensor does not need any driving structures and any additional driving signals. Therefore, this sensor does not need a complex excitation circuit, and it has lower power consumption. In the revised manuscript, we compare with other MEMS voltage sensors. Moreover, the test circuit of the sensor is simple and the static power consumption of the circuit depends on the series resistor R0 in the manuscript. And when the series resistor R0 is 100kΩ, the circuit power consumption is on the order of mW or lower.
|
Source |
Driving structure |
The driving voltage |
|
[22] |
thermal drive |
driving voltage 75mV(vacuum environment) |
|
[23] |
electrostatic drive |
DC bias voltage 20V and AC voltage amplitude 1V |
|
[24] |
electrostatic drive |
DC bias voltage 8V and AC voltage 10V(peak to peak) |
|
This work |
no driving structure |
no driving voltage |
[22] Wijeweera G , Bahreyni B , Shafai C , et al. Micromachined Electric-Field Sensor to Measure AC and DC Fields in Power Systems[J]. IEEE Transactions on Power Delivery, 2009, 24(3):988-995.
[23] Yang P , Wen X , Yao L Chu Z ,Peng C. A non-intrusive voltage measurement scheme based on MEMS electric field sensors: Theoretical analysis and experimental verification of AC power lines, Review of Scientific Instruments, 2021,92,065002.
[24] Mm A , Km B , App A . MEMS-based non-contact voltage sensor with multi-mode resonance shutter[J]. Sensors and Actuators A: Physical, 2019, 294:25-36.

Reviewer 2 Report
The authors proposed a novel piezoresistive MEMS voltage sensor. The manuscript is well written and well-illustrated. However, those are my queries:
- "The sensor has demonstrated that it can measure the voltage of 1V from 50Hz to 22 300Hz and 0.5V from 400Hz to 1000Hz. In the voltage measurement range of 900V-1200V at the 23 power frequency of 50Hz, the linearity of the sensor is 3.4%, and 0.93% in the voltage measurement 24 range of 400V-1200V at the frequency of 200Hz.". This excerpt is confusing. I think the authors mean that the lowest voltage amplitude that could be accurately measured is 1V. The authors should reformulate the text to avoid misunderstandings.
- The authors used good references. However, more recent studies should be included because many of the references are before 2017.
- Please, enlarge Figure 10.
- The authors should point out some future works in the conclusion section.
- The introduction must present more background information and a broader literature review.
Author Response
Dear Reviewer,
Thanks for your valuable comments for the manuscript entitled “Design and Testing of a Novel Non-contact MEMS Voltage Sensor Based on Single-Crystal Silicon Piezoresistive Effect”. We have tried our best to revise the manuscript. Revised portions are marked using the “Track Changes” function in the revised manuscript. And the point to point responds to your comments are listed as following:
Point 1: "The sensor has demonstrated that it can measure the voltage of 1V from 50Hz to 300Hz and 0.5V from 400Hz to 1000Hz. In the voltage measurement range of 900V-1200V at the power frequency of 50Hz, the linearity of the sensor is 3.4%, and 0.93% in the voltage measurement range of 400V-1200V at the frequency of 200Hz.". This excerpt is confusing. I think the authors mean that the lowest voltage amplitude that could be accurately measured is 1V. The authors should reformulate the text to avoid misunderstandings.
Response 1: Thank you very much for your suggestion. The sentence ”The sensor has demonstrated that it can measure the voltage of 1V from 50Hz to 300Hz and 0.5V from 400Hz to 1000Hz.” is rewritten as “The sensor has demonstrated that the minimum detectable voltage is 1V for a 50Hz ac voltage to a 300Hz ac voltage, and 0.5V for a 400Hz ac voltage to a 1000Hz ac voltage.”.
Point 2: The authors used good references. However, more recent studies should be included because many of the references are before 2017.
Response 2: Thank you very much for your suggestion. In the revised manuscript, we have added some recent studies after 2017.
Point 3: Please, enlarge Figure 10.
Response 3: Thank you very much for your suggestion. In the revised manuscript, we have enlarged Figure 10, as shown below:
Point 4: The authors should point out some future works in the conclusion section.
Response 4: Thank you very much for your suggestion. In the revised manuscript, we have pointed out some future works in the conclusion section. That is “In the future, we will further carry out application verification for practical application scenarios such as power transmission, transformation and distribution, and study the anti-interference characteristics of sensors. In addition, for low-voltage measurements, such as below 220V, further optimize the sensor to improve the measurement sensitivity.”
Point 5: The introduction must present more background information and a broader literature review.
Response 5: Thank you very much for your suggestion. In the revised manuscript, we have added some recent studies about non-contact voltage sensors based on optical fiber and coupling capacitor to widen the literature review.

Round 2
Reviewer 1 Report
I think the novelty of this work is not high, so it is not appropriate to use the word of "novel" in the title of this work.
Author Response
Dear Reviewer,
Thanks for your valuable comments for the manuscript entitled “Design and Testing of a Novel Non-contact MEMS Voltage Sensor Based on Single-Crystal Silicon Piezoresistive Effect”. We have tried our best to revise the manuscript. Revised portions are marked using the “Track Changes” function in the revised manuscript. And the point to point responds to your comments are listed as following:
Point 1: I think the novelty of this work is not high, so it is not appropriate to use the word of "novel" in the title of this work.
Response 1: Thank you very much for your suggestion. The purpose of this paper is to propose a MEMS voltage sensor based on a single-crystal silicon piezoresistive beam structure and to verify its output response and test performance. Compared with conventional piezoresistive sensors such as ion implantation, doping, or diffusion, this paper uses the entire single-crystal silicon beam as the piezoresistive element, which simplifies the fabrication process, improves the temperature characteristics, and increases the sensitivity. To avoid misunderstanding and combining your comments, I also change the title of the paper to "Design and Testing of a Non-contact MEMS Voltage Sensor Based on Single-Crystal Silicon Piezoresistive Effect ".
